# Evaluating Influenza Vaccination Practices among COPD Patients

**DOI:** 10.3390/vaccines12010014

**Published:** 2023-12-22

**Authors:** Walid Al-Qerem, Anan Jarab, Judith Eberhardt, Fawaz Alasmari, Saja K. AbedAlqader

**Affiliations:** 1Department of Pharmacy, Faculty of Pharmacy, Al-Zaytoonah University of Jordan, Amman 11733, Jordan; waleed.qirim@zuj.edu.jo (W.A.-Q.); 201911073@zuj.edu.jo (S.K.A.); 2College of Pharmacy, Al Ain University, Abu Dhabi 112612, United Arab Emirates; anan.jarab@aau.ac.ae; 3AAU Health and Biomedical Research Center, Al Ain University, Abu Dhabi 112612, United Arab Emirates; 4Department of Clinical Pharmacy, Faculty of Pharmacy, Jordan University of Science and Technology, Irbid 22110, Jordan; 5Department of Psychology, School of Social Sciences, Humanities and Law, Teesside University, Borough Road, Middlesbrough TS1 3BX, UK; 6Department of Pharmacology and Toxicology, College of Pharmacy, King Saud University, Riyadh 12372, Saudi Arabia; ffalasmari@ksu.edu.sa

**Keywords:** influenza, COPD, morbidity, complications, influenza vaccine

## Abstract

Chronic Obstructive Pulmonary Disease (COPD) stands as a global health concern linked to considerable morbidity and mortality. In Jordan, the prevalence of COPD is substantial, but research in this area is limited. Exacerbations of COPD can lead to severe outcomes, including hospitalization and increased cardiovascular risk. Influenza is a significant trigger of exacerbations in COPD patients, and vaccination is recommended. However, studies have shown negative attitudes towards the influenza vaccine. This cross-sectional study aimed to investigate the knowledge, attitudes, practices, and intentions of COPD patients in Jordan regarding influenza vaccination. Data were collected through a custom-designed questionnaire from 300 COPD patients. The study revealed low influenza vaccination rates, with forgetfulness and lack of knowledge about vaccine effectiveness being the main barriers. Higher knowledge and positive attitudes were associated with greater intention to vaccinate. To tackle these challenges, it is recommended to implement customized health education campaigns, foster collaborations with healthcare providers, and engage in community-focused initiatives to enhance acceptance of the influenza vaccine among COPD patients in Jordan. These findings underscore the importance of addressing knowledge gaps and negative attitudes to enhance vaccine uptake and improve health outcomes for COPD patients.

## 1. Introduction

COPD is defined as chronic airflow obstruction that is fatal if not properly managed and remains underdiagnosed. It constitutes a major contributor to global mortality and morbidity. One of the leading causes of COPD is smoking, and according to the World Health Organization (WHO), the prevalence of Jordanian smokers has reached 54.9% among men above 18 years old [1,2]. However, the prevalence of COPD in Jordan is not well studied; a recent study found that the prevalence of COPD in smoking male adults was 48.2% [3]. The prevalence rate of COPD in Jordan is 6.5% in patients under 50 and 37.5% in patients over 70, which is significantly higher than the reported prevalence rate worldwide, particularly for the older age group [4].

The main aims of COPD treatment lie in reducing symptoms, improving patients’ quality of life, and minimizing exacerbations that can progress to a pulmonary function decline. Furthermore, these exacerbations can elevate the risk of cardiovascular diseases, occurring independently of the degree of pulmonary function impairment. This ultimately heightens the likelihood of hospitalization for COPD patients [5,6].

Upper respiratory tract viral infections are primary triggers for acute-severe COPD exacerbations. Based on the most recent information released by the World Health Organization (WHO) in 2020, 3.59% of the overall fatalities in Jordan were attributed to influenza and pneumonia [7]. Thankfully, influenza can be prevented through vaccination. The Centers for Disease Control and Prevention (CDC) in the United States advises that individuals at high risk, aged six months and older, are vaccinated against influenza yearly. This vaccination is recommended during the period from September to the end of October, preceding the typical seasonal influenza period [8]. Moreover, the Global Initiative for Chronic Obstructive Lung Disease (GOLD) advocates for influenza vaccination among individuals with COPD. Nevertheless, previous research reported that individuals with COPD exhibited the lowest influenza vaccination coverage rate when compared to other high-risk groups, such as those with diabetes and those undergoing dialysis [9]. A 2019 study concluded that older adults in Jordan exhibited a negative attitude toward the vaccine [10]. Additionally, during the COVID-19 pandemic, Jordanians were found to have low intentions of receiving the influenza vaccine, reaching only 27.7% [11].

COPD patients are at a heightened risk of complications arising from infection with influenza due to various factors. These include chronic inflammation of the airways and compromised innate immune cell responses, which contribute to severe influenza infection, exacerbation of COPD symptoms, and an increased chance of developing pneumonia [12,13]. Given that COPD heightens the risk of severe infection with influenza, prioritizing vaccination for these patients is imperative, irrespective of potential side effects [14]. Despite the absence of studies investigating knowledge, attitudes, practices, and intentions associated with influenza vaccination in COPD patients in Jordan and the region, the findings of the present study can contribute to assisting healthcare providers in crafting health campaigns. These campaigns could promote influenza vaccination among COPD patients, ultimately enhancing the control of COPD.

Hence, this study examined COPD patients’ understanding of COPD, influenza, and its vaccine. It also focused on examining adherence to COPD practices aimed at enhancing control, as well as exploring attitudes and practices associated with influenza vaccination. Additionally, the study sought to identify variables linked to the intention of receiving the vaccine in the current year and to investigate reasons behind non-vaccination.

## 2. Materials and Methods

The current study was conducted with COPD patients who attended respiratory outpatient clinics at Albasheer Hospital in the period between January and June 2023. The criteria for inclusion encompassed adults aged 18 years and over, diagnosed with COPD for a minimum of one year, and expressing willingness to take part. The files of COPD patients who had a regular follow-up appointment the next day were reviewed and those who fulfilled the inclusion criteria were approached when they arrived at the clinic. Each patient received a concise overview of the study’s objectives before the research pharmacist conducted interviews with them. Participants were briefed on the principles of confidentiality, data anonymity, and voluntary participation. Subsequently, all participants signed an informed consent form. The study followed the Declaration of Helsinki’s ethical guidelines. Ethical approval was secured from Al-Zaytoonah University of Jordan (Ref#18/09/2022–2023).

### 2.1. Data Collection and Study Instruments

Data collection utilized a questionnaire custom-designed on Google Forms. This instrument was crafted following a thorough literature review and subjected to back translation into Arabic by two translators working independently. The questionnaire comprised six sections. The initial section focused on assessing the medical and demographic profile of participants, encompassing sex, age, marital and socioeconomic status, education, smoking habits, being exposed to passive smoking, COPD duration, and any hospitalization as a result of COPD in the preceding year. Section two included the COPD assessment test (CAT), which was developed by Jones et al. and consisted of eight questions; the questionnaire was deemed reliable (Cronbach’s alpha = 0.88) [15]. Higher scores indicated a greater impact of COPD on a patient’s life. In the third section, patients’ understanding of COPD, influenza, and the influenza vaccine was assessed, with each correct response earning a single point. The total score, representing the sum of points, had a maximum possible value of 17, with higher scores indicating greater knowledge. In the fourth section, the questionnaire gauged patients’ attitudes toward influenza vaccination using a Likert scale with four items (‘strongly disagree’, ‘disagree’, ‘neutral’, ‘agree’, or ‘strongly agree’), ranging from one point for ‘strongly disagree’ to five points for ‘strongly agree’. The negatively worded question was reverse coded. The overall score was computed by summing all points, with a maximum achievable score of 20. A greater score suggested a more positive attitude towards the vaccine. The fifth part evaluated COPD practices, which consisted of four items, with a Likert scale (‘never’, ‘rarely’, ‘sometimes’, ‘usually’, and ‘always’) ranging from one point for ‘never’ to five points for ‘always’. The total score was computed by adding up all points, with a maximum achievable score of 20. The sixth part of the questionnaire included the perceived dangerousness of the influenza vaccine, prior experiences with side effects, their severity, and influenza vaccination intention for the current year. The concluding section focused on identifying barriers faced by respondents who had not received the annual influenza vaccine, did not plan on receiving it in the present year, or were uncertain about their vaccination intentions. The questionnaire is included in the Appendix A.

### 2.2. Sample Size Calculation

A minimum required sample size of 273 participants was determined using a convenience sampling technique, with a 50% population proportion, a 90% significance level, and a 5% error margin. The current study included 300 patients [16].

### 2.3. Tool Validation

To establish content validity, an expert panel consisting of two pulmonologists and a clinical pharmacist was consulted. A pilot study, involving 27 COPD patients, was conducted to confirm the clarity of the questionnaire and its comprehensibility for Jordanian participants. The data from the pilot study were not included in the final analysis. Furthermore, principal component analysis (PCA) was conducted to ensure the validity of the attitude towards influenza vaccine and COPD practices scales. Additionally, to ascertain the reliability of the results, Cronbach’s alpha values were calculated to evaluate the internal consistency of the computed scores (CAT, knowledge about COPD, influenza, and the vaccine, attitudes toward the vaccine, and COPD practices). Acceptable values are 0.7 or above.

### 2.4. Analysis

The data were analyzed using SPSS version 26. Continuous variables’ normality was assessed using Q-Q plots and Kolmogorov–Smirnov tests. The results indicated a non-normal distribution for the continuous variables. PCA was conducted to evaluate the validity of the attitude towards the influenza vaccine and COPD practice scales. The suitability of the data for conducting PCA was determined through Kaiser–Meyer–Olkin (KMO) analysis and Bartlett’s Test of Sphericity. The suitable number of factors to extract was determined by conducting parallel analysis. A pattern matrix was generated using promax rotation. Communalities were evaluated and any item with a communality below 0.3 was removed from the analysis; the factors loadings were evaluated and any item with loadings below 0.4 or with multiple loadings above 0.4 was dropped from the analysis. To investigate the relationships between variables and the inclination to get vaccinated against influenza in the present year, bivariate analyses were conducted, employing Chi-square tests for categorical variables and Mann–Whitney U tests for continuous variables. For multivariate analysis, a multinomial logistic regression model was developed, incorporating independent variables with *p*-values below 0.2 from the bivariate analysis. The significance threshold for all analyses was set at *p* < 0.05.

## 3. Results

The sample comprised 300 COPD patients, with a median age of 55 years (range 54–58); 52.3% were women. Most (84.7%) were married, and 54.3% of the patients were earning JOD 500 or more per month (USD 705). The median duration of COPD diagnosis was 4 years (range 4–5), and 80.0% of patients had experienced hospitalization in the previous year due to COPD. Additionally, 32.7% of the patients were smokers. The median CAT score was 32 (32–33) with a maximum score of 40 (Cronbach’s alpha for CAT = 0.73). Men in the present sample were older (58 vs. 54), less exposed to passive smoking (38.5% vs. 73.9%), and had a higher frequency of current smoking (49% vs. 17.8%) than women. All other variables were comparable between the two sexes (Table 1).

Table 2 summarizes patient responses to knowledge items pertaining to COPD, influenza, and the vaccine. The comprehensive knowledge score’s median was 6 (range 6–7) with a maximum possible score of 17 (Cronbach’s alpha = 0.72). Notably, the item exhibiting the most correct answers in the ‘COPD knowledge’ category was “Patients may experience flare-ups or exacerbations” (78%) followed by “COPD is an infectious disease” and “Does your psychological state affect COPD symptoms?” (67.3% and 67%, respectively). Conversely, the lowest values were obtained for the item “Do you know the spirometry test?”, with only 14.3% responding with an affirmative. Concerning ‘influenza knowledge,’ 70% of the patients provided correct responses to “Influenza can spread from one person to another”, and just 14.7% of the patients correctly responded to “Antibiotics can be used to treat flu”. Finally, the top score in the ‘influenza vaccine knowledge’ part was achieved on the item “Is there a vaccine against the flu?” (54.3%), whilst the fewest correct answers were given on the item “Does the vaccine have side effects?” (6.7%).

Table 3 displays patients’ responses to items assessing attitudes toward the influenza vaccine. KMO and Bartlett’s test for sphericity confirmed the data’s suitability for conducting PCA (KMO = 0.73, *p* < 0.001). The PCA produced a one-factor solution as suggested by parallel analysis (Appendix A in the Appendix A) with communalities ranging from 0.31 to 0.717 and factor loadings ranging from 0.487 to 0.847 with a Cronbach’s alpha value of 0.71. The attitude score’s median was 12 (range 11–13) out of a maximum possible score of 20. The highest percentage of patients ‘strongly agreed/agreed’ with the item “Influenza vaccination prevents infection by the influenza virus” (26.3%), while the lowest percentage of patients (19.9%) ‘strongly agreed/agreed’ with the statement “My physician believes that I should receive the influenza vaccine”. On the reverse-coded statement, 17.4% of the patients ‘strongly disagreed/disagreed’ with “I believe that I can get sick because of the influenza shot”.

COPD practices were evaluated based on four statements, as illustrated in Table 4. KMO and Bartlett’s test for sphericity confirmed the suitability of the data for conducting PCA (KMO = 0.72, *p* < 0.001). The PCA produced a two-factor solution as suggested by parallel analysis (Appendix A in the Appendix A) with communalities ranging from 0.728 to 0.867 and factor loadings ranging from 0.838 to 0.954 with Cronbach’s alpha values of 0.848 and 0.72. The median score for COPD practices was 9 (8–11) out of a maximum attainable score of 20. The practice reported most frequently among COPD patients consisted of the item “How often do you avoid exposure to dust/air pollution?” with 29.4% of patients reporting such practices as “always/usually”. In contrast, the least frequently reported practice among the patients was “How regularly do you engage in physical activities?” (7.6%).

The responses of COPD patients to the items assessing influenza vaccine practices are detailed in Table 5. A significant majority of patients (80.3%) had never received the influenza vaccine. Headache was the most frequently reported side effect, with 40.6%, and redness was reported least, accounting for 15.3%. Moreover, 67.8% of vaccinated patients reported experiencing mild side effects following vaccination. Regarding their intention to get vaccinated against influenza in the present year, 13.7% confirmed their intent to do so, while 51.3% were uncertain.

Bivariate analysis, employing Chi-square and Mann–Whitney U tests, examined the relationship between the intention to get vaccinated against influenza in the present year and various sociodemographic variables including knowledge, attitudes, COPD practices, age, education, income, sex, smoking habits, CAT, marital status, COPD duration, and hospitalization due to COPD. The education level (*p* = 0.013), knowledge score (*p* < 0.001), COPD practice (*p* = 0.042), and attitude score (*p* < 0.001) exhibited significant associations with answering yes vs. no to the question “Do you intend to take the influenza vaccine this year?”, while between those who answered maybe vs. no, the significant variables were education level (*p* = 0.003), knowledge score (*p* < 0.001), and attitude score (*p* < 0.001).

A multinomial logistic regression analysis was conducted to determine the relationship between sociodemographic factors and influenza vaccination intention for the present year (Table 6). Included independent variables were those with *p*-values less than 0.2 in the bivariate analysis: knowledge, attitudes, COPD practices, age, education, and hospitalization due to COPD. The analysis indicated that with an increase in the knowledge score, there was a decrease in the likelihood of rejecting the influenza vaccine (OR = 0.778, 95% CI [0.633–0.957], *p* = 0.017). Additionally, an increase in the attitude score was associated with a decrease in both rejection and hesitancy towards getting vaccinated (OR = 0.263, 95% CI [0.191–0.363], *p* < 0.001, and OR = 0.497, 95% CI [0.388–0.636], *p* < 0.001, respectively).

Figure 1 presents reasons for not wishing to receive the influenza vaccine. The predominant explanation for not getting vaccinated was forgetfulness (36%), followed by doubting its effectiveness (31.5%). Conversely, the least-frequently reported factor was cost (4.5%).

## 4. Discussion

The influenza vaccine coverage rate in Jordan is remarkably low, standing at only 9.9% among Jordanian adults during the 2011/2012 season [17]. This is also evident in studies reporting low acceptance of the influenza vaccine among high-risk groups in Jordan, such as patients with asthma or diabetes mellitus [18,19,20]. However, healthcare workers exhibited a substantially higher vaccination rate of 50% [17]. Vaccination is proven to be an important preventive healthcare measure that enhances health outcomes and prevents infection-induced COPD exacerbation. Even with ongoing recommendations to vaccinate COPD patients, vaccination rates continue to fall below optimal levels in this patient group [21]. Furthermore, the diversity in the factors associated with decreased willingness to get vaccinated necessitates further research to help pinpoint these factors and develop interventions accordingly.

A low knowledge score in relation to COPD, influenza, and its vaccine was observed in the present study. The observed deficiency in knowledge regarding influenza and its vaccine is consistent with previous studies conducted in Jordan, particularly among patients with chronic conditions [22] and asthmatic patients [18]. Nevertheless, contrasting findings emerge from research conducted in Saudi Arabia [23], China [24], and South Africa [25] which reported a high level of knowledge about influenza and its vaccine. This highlights the need for educational approaches to enhance COPD patients’ understanding of COPD, influenza, and influenza vaccination.

An average attitude score in relation to the influenza vaccine was observed among COPD patients. Nevertheless, 27.4% of patients held the belief that they might fall ill due to the influenza vaccine, and 12.2% mentioned that their physician had not recommended getting the influenza vaccine. The absence of physician recommendations and the existence of misconceptions about the influenza virus and its vaccine could potentially impact attitudes toward the vaccine negatively. Previous studies have highlighted that doctors’ recommendations play a crucial role in motivating patients to receive the influenza vaccine [18,19,26]. This highlights the importance of educating patients about the benefits of being vaccinated and how it contributes to preventing disease complications and improving health outcomes. It is deemed necessary to implement educational interventions that motivate physicians to convey the necessity of vaccination for individuals with COPD.

The influenza vaccination rate among COPD patients in this study was notably low, as only 19.7% reported ever receiving the influenza vaccine, and a mere 7% indicated an annual vaccination routine. Furthermore, merely 13.7% intended to get vaccinated in the present year. These results align with a study conducted in Jordan among asthmatic children, revealing that 39.6% had been vaccinated for influenza at any point, and only 10.5% received it annually [18]. In another study conducted in Turkey, the rate of regular annual vaccination was found to be only 10.3% in the vaccinated group [26]. A pilot study showed that by conducting routine audits to monitor progress and employing a range of simple interventions to implement, enhancing rates of influenza vaccination is a feasible objective [27].

Forgetfulness (36%) and a lack of awareness about the vaccine’s effectiveness (31.5%) were the most cited reasons for not getting vaccinated. This underscores the necessity for focused educational initiatives to enhance awareness regarding the importance of influenza vaccination for individuals with COPD. Previous studies have demonstrated that interventions initiated by pharmacists effectively enhance the adoption of influenza vaccines in patients with COPD or asthma [28]. Moreover, tackling concerns related to forgetfulness and devising strategies to remind COPD patients to undergo vaccination, such as the use of text message reminders, has proven to be a successful approach for increasing influenza vaccine uptake [29].

The multinomial regression model revealed that knowledge about COPD, influenza, and its vaccine, along with attitudes toward the vaccine, were linked to the intention to vaccinate against influenza. Elevated knowledge and attitude scores were correlated with a decrease in influenza vaccine hesitancy and rejection. The reason for this connection lies in the fact that such individuals tend to place higher importance on their health, have a better understanding of the advantages of vaccination, and feel motivated to take preventive measures to avoid contracting influenza. Insufficient knowledge has consistently been recognized as a hurdle to influenza vaccination, both among at-risk populations and the public [30]. This underlines the significance of education/information provision to improve influenza vaccination rates among patients with COPD in Jordan.

Implementing community-focused initiatives, such as campaigns that are culturally sensitive, as well as outreach programs, can contribute to ensuring equitable access to vaccinations, particularly in deprived areas. Partnering with health professionals to strengthen educational initiatives can foster trust and alleviate concerns related to costs. Additionally, sustained educational campaigns led by government agencies, regional health departments, medical professionals, and community organizations can help counter misconceptions and underscore the significance of vaccination for overall health. Another potentially effective strategy involves designing culturally tailored health education campaigns to dispel misconceptions. Moreover, cultivating enhanced collaboration between healthcare providers and public health experts would guarantee the dissemination of accurate information. These strategies could help public health practitioners counter misinformation, raise awareness, and boost influenza vaccination rates among COPD patients in Jordan.

### Study Limitations

The present study is subject to certain limitations. The data collection depended on self-reported measures, introducing the possibility of social desirability and recall biases. However, the low vaccination rates evidenced in this study contribute to the robustness of the results. While the study site serves a significant patient population across a broad geographic region, including additional healthcare facilities or clinics would improve the generalizability of the results. Additionally, the current sample size was calculated to produce a 90% confidence level. While this remains a high confidence level, it would have been more appropriate to increase the sample size in order to reach a 95% confidence level. Although the present study evaluated several variables that may influence vaccination habits, other variables that may have influenced these habits were not assessed in the present study, including adherence levels. Given that participation in the current study was voluntary, this may have increased the risk for self-selection bias, as patients with higher health literacy are more likely to participate in health research [31]; however, the variability presented in the study sample in terms of education, age group, and income level minimizes the possible impact of a selection bias, which is also confirmed by the low levels of knowledge, negative attitudes, and low acceptance rates of influenza vaccination reported in the present study. Finally, it is worth noting that we did not gather qualitative data, which may have offered a more profound insight into the beliefs and attitudes of COPD patients regarding both COPD and influenza vaccination. Future research should endeavor to collect data from a variety of sites, combining quantitative and qualitative methods to provide more in-depth insights into the factors driving influenza vaccine uptake among COPD patients.

## 5. Conclusions

This study revealed low influenza vaccination rates among patients with COPD. Lower influenza vaccination intentions were associated with decreased knowledge and negative attitudes toward the vaccination process. These findings emphasize the urgent need for implementing interventions that tackle the knowledge gaps and the negative attitudes identified in the present study to enhance the intention to receive the influenza vaccine and consequently improve health outcomes for individuals with COPD.

## Figures and Tables

**Figure 1 vaccines-12-00014-f001:**
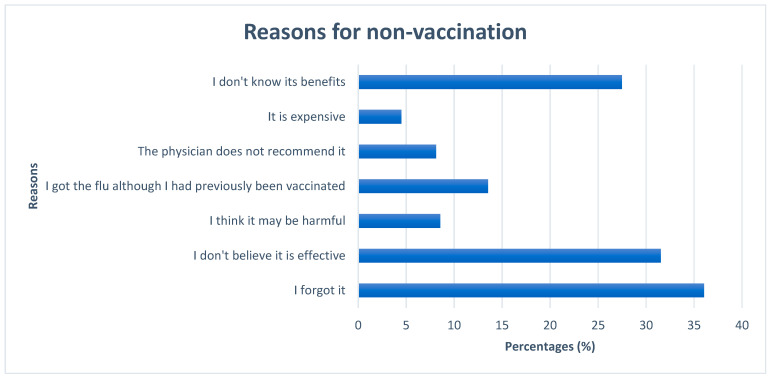
Patients’ reasons for non-vaccination.

**Table 1 vaccines-12-00014-t001:** Patients’ sociodemographic characteristics.

		Men N = 143 (47.7%)	Women N = 157 (52.3%)	Total N = 300
		Median (95%Cl) or Frequency (%)	Median (95%Cl) or Frequency (%)	Median (95%Cl) or Frequency (%)
Age		58 (55–62)	54 (52–57)	55 (54–58)
Educational level	Elementary/secondary	96 (67.1%)	107 (68.2%)	203 (67.7%)
	High school or higher	47 (32.9%)	50 (31.8%)	97 (32.3%)
Marital status	Married	121 (84.6%)	133 (84.7%)	254 (84.7%)
	Non currently married	22 (15.4%)	24 (15.3%)	46 (15.3%)
Monthly income	Less than JOD 500	59 (41.3%)	73 (46.5%)	132 (44.0%)
	JOD 500 or more	84 (58.7%)	84 (53.5%)	168 (56%)
COPD duration		4 (4–5)	4 (4–5)	4 (4–5)
Previous COPD hospitalization	No	31 (21.7%)	29 (18.5%)	60 (20.0%)
	Yes	112 (78.3%)	128 (81.5%)	240 (80.0%)
Smoking status	Former smoker	73 (51%)	129 (82.2%)	202 (67.3%)
	Smoker	70 (49%)	28 (17.8%)	98 (32.7%)
Are you exposed to passive smoking?	No	78 (54.5%)	39 (24.8%)	117 (39.0%)
	I don’t know	10 (7%)	2 (1.3%)	12 (4.0%)
	Yes	55 (38.5%)	116 (73.9%)	171 (57.0%)
COPD Assessment Test (CAT)		31 (30–33)	32(32–34)	32 (32–33)

**Table 2 vaccines-12-00014-t002:** Patients’ answers to knowledge questions about COPD, influenza, and vaccination.

		Median (95%Cl) or Frequency (%)
COPD Knowledge		
COPD is an infectious disease	No *	202 (67.3%)
	I don’t know	96 (32%)
	Yes	2 (0.7%)
Can genetics play a role in a person’s susceptibility to COPD?	No	138 (46%)
	I don’t know	104 (34.7%)
	Yes *	58 (19.3%)
Patients may experience flare-ups/exacerbations	No	19 (6.3%)
	I don’t know	47 (15.7%)
	Yes *	234 (78%)
Do you know the spirometry test?	No	142 (47.3%)
	I don’t know	115 (38.3%)
	Yes *	43 (14.3%)
Does the flu make the symptoms of COPD worse?	No	56 (18.7%)
	I don’t know	80 (26.7%)
	Yes *	164 (54.7%)
Do you know the benefits of using COPD inhalers?	No	59 (19.7%)
	I don’t know	116 (38.7%)
	Yes *	125 (41.7%)
Do you know how to correctly use COPD inhalers?	No	36 (12%)
	I don’t know	104 (34.7%)
	Yes *	160 (53.3%)
Engaging in physical activities helps improve COPD symptoms.	No	177 (59%)
	I don’t know	63 (21%)
	Yes *	60 (20%)
Does diet affect COPD symptoms?	No	152 (50.7%)
	I don’t know	55 (18.3%)
	Yes *	93 (31%)
Does your psychological state affect COPD symptoms?	No	48 (16%)
	I don’t know	51 (17%)
	Yes *	201 (67%)
Influenza knowledge		
Influenza is the same as the common cold	No	62 (20.7%)
	I don’t know	187 (62.3%)
	Yes *	51 (17%)
Influenza is caused by bacteria	No *	53 (17.7%)
	I don’t know	219 (73%)
	Yes	28 (9.3%)
Influenza can spread from one person to another	No	1 (0.3%)
	I don’t know	89 (29.7%)
	Yes *	210 (70%)
Antibiotics can be used to treat flu	No *	44 (14.7%)
	I don’t know	117 (39%)
	Yes	139 (46.3%)
Influenza vaccine knowledge		
Is there a vaccine against the flu?	No	18 (6%)
	I don’t know	119 (39.7%)
	Yes *	163 (54.3%)
Does the vaccine have side effects?	No	30 (10%)
	I don’t know	250 (83.3%)
	Yes *	20 (6.7%)
When is the appropriate time to take the influenza vaccine?	1–3 (January–March)	13 (4.3%)
	11–12 (November–December)	20 (6.7%)
	9–10 (September–October) *	25 (8.3%)
	I don’t know	18 (6%)

* Represents the correct answer.

**Table 3 vaccines-12-00014-t003:** Patients’ responses to items assessing attitudes toward the influenza vaccine.

	Strongly Disagree	Disagree	Neutral	Agree	Strongly Agree
	Frequency (%)	Frequency (%)	Frequency (%)	Frequency (%)	Frequency (%)
I believe that I must receive the influenza vaccination	5 (1.7%)	49 (16.3%)	178 (59.3%)	52 (17.3%)	16 (5.3%)
My physician believes that I should receive the influenza vaccine	7 (2.4%)	29 (9.8%)	202 (68%)	40 (13.5%)	19 (6.4%)
Influenza vaccination prevents infection by the influenza virus	3 (1%)	30 (10.1%)	182 (61.3%)	59 (19.9%)	23 (6.4%)
I believe that I can get sick because of the influenza shot *	3 (1%)	49 (16.4%)	166 (55.3%)	68 (22.7%)	14 (4.7%)

* Reverse-coded statement.

**Table 4 vaccines-12-00014-t004:** Patients’ responses to COPD practice items.

	Never	Rarely	Sometimes	Usually	Always
	Frequency (%)	Frequency (%)	Frequency (%)	Frequency (%)	Frequency (%)
How often do you avoid smoking/exposure to smoking?	65 (21.7%)	75 (25%)	89 (29.7%)	51 (17%)	20 (6.7%)
How often do you avoid exposure to dust/air pollution?	24 (8%)	79 (26.3%)	109 (36.3%)	71 (23.7%)	17 (5.7%)
How regularly do you engage in physical activities?	150 (50%)	90 (30%)	37 (12.3%)	13 (4.3%)	10 (3.3%)
How closely do you follow a healthy diet?	67 (22.3%)	121 (40.3%)	57 (19%)	36 (12%)	19 (6.3%)

**Table 5 vaccines-12-00014-t005:** Patients’ responses to items assessing influenza vaccine practices.

		Median (95%Cl) or Frequency (%)
In your opinion, how dangerous is the flu to your health?		3 (3–4)
Do you intend to take the influenza vaccine this year?	No	105 (35%)
	Not sure	154 (51.3%)
	Yes	41 (13.7%)
How often did you receive the influenza vaccine?	Never	241 (80.3%)
	Once	26 (8.7%)
	More than once	12 (4%)
	Annually	21 (7%)
What was the severity of the side effects?	Mild	40 (67.8%)
	Moderate	16 (27.1%)
	Severe	3 (5.1%)
Fever		14 (23.7%)
Redness		6 (10.2%)
Fatigue		9 (15.3%)
Headache		24 (40.6%)
Nausea		11 (18.6%)

**Table 6 vaccines-12-00014-t006:** Multinomial regression model examining the relationship between sociodemographic characteristics and the intention to receive influenza vaccination in the current year.

		*p*-Value	OR	95% Confidence Interval for OR	
				Lower Bound	Upper Bound
No vs. yes	Intercept	<0.001			
	Knowledge score	0.017	0.778	0.633	0.957
	Attitude score	<0.001	0.263	0.191	0.363
	COPD practice score	0.445	0.937	0.793	1.107
	Age	0.642	1.012	0.963	1.063
	Education (low vs. high)	0.997	1.002	0.324	3.095
	Hospitalized (no vs. yes)	0.832	0.849	0.195	3.862
Not sure vs. yes	Intercept	<0.001			
	Knowledge score	0.090	0.852	0.708	1.025
	Attitude score	<0.001	0.497	0.388	0.636
	COPD practice score	0.382	0.937	0.806	1.088
	Age	0.664	0.990	0.947	1.036
	Education (low vs. high)	0.244	1.768	0.678	4.610
	Hospitalized (no vs. yes)	0.833	1.200	0.288	4.691

## Data Availability

Available upon request.

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
