# Peer review of "Evaluating Influenza Vaccination Practices among COPD Patients"

_vaccines, 2023, doi:10.3390/vaccines12010014_

Round 1

Reviewer 1 Report

Comments and Suggestions for Authors

This study evaluates the COPD patients’ knowledge about COPD, influenza and its vaccine, adherence to COPD practices that improve control, and attitudes and practices towards vaccinating against influenza in Jordan. This article is of interest to the health administration, but some minor revisions are necessary.

 1.- I would try to modify some parts of the title, it is quite similar to the one already published by the authors for young adults with asthma. “Al-Qerem W, Alassi A, Jarab A, Al Bawab AQ, Hammad A, Alasmari F, Alazab B, Abu Husein D, Al Momani N, Eberhardt J. Examining Influenza Vaccination Patterns Among Young Adults with Asthma: Insights into Knowledge, Attitudes, and Practices. Patient Prefer Adherence. 2023 Nov 10;17:2899-2913.”

 2.- In the introduction they say: "The US Centers for Disease Control (CDC) recommends 48 that all people aged six months and older be vaccinated against influenza every year, from 49 September until the end of October, before the usual period of seasonal influenza". This sentence is not correct, influenza vaccination is not recommended for all people over six months of age (only if they have any comorbidity). The reference used [7] is the indication for influenza vaccination for people with asthma.

3.- In the methods section they say: "The current study was conducted with COPD patients who attended outpatient clinics at Albasheer Hospital". Could you specify in more detail how COPD patients have been selected? It seems incongruous that they select outpatients in a hospital.

 4.- In Figure 1 it would be desirable to fully show the possible answers, since, although some are mention in the main text, we have no other way to know them [maybe an horizontal histogram (vs vertical histogram) would be more appropriate to achieve that].

 5.- One item on which I need clarification is the “smoking status” in the first section of the questionnaire. The two possible answers shown are “Smoker” and “Former smoker”: does that mean that never smokers are not considered in the study or they are included as “Former smokers”?

Reviewer 2 Report

Comments and Suggestions for Authors

The study aims to understand the attitude of COPD patients towards influenza vaccination.

1.       The authors state that the questionnaire “was translated back to Arabic by two independent translators” (line 84). Was there perhaps an original written in another language?

2.       The questionnaire included the COPD Assessment Test (CAT), a questionnaire for people with COPD. It is designed to measure the impact of COPD on a person's life, and how it changes over time. The Authors should indicate the author of the questionnaire and the Cronbach's alpha they obtained in this study.

3.       The 3rd, 4th , 5th and 6th parts of the questionnaire were produced ad hoc and were not validated. The authors should describe these instruments in detail, make it known how the questions were constructed and what the scores were.

4.       The authors declare that they used an internal consistency test (Cronbach's alpha). As is known, this test is not sufficient to validate a questionnaire. It is recommended to perform at least a principal component analysis.

5.       The authors state that the questionnaire has been verified by experts. However, they did not realize that at least one of the questions contains the correct answer. Indeed, the item “Do you perform lung function testing at every doctor's visit?” cannot have a frequency as an answer, because the frequency is already expressed in the question.

6.       To calculate the sample size, the authors chose a significance level of 90%. They should have correctly chosen a 95% level, considering significance at p<0.05. The correct calculation indicates that at least 385 cases are needed to estimate the responses with a margin of error of 5%

7.       The authors should indicate the computerized sample size calculator they used.

8.       In the logistic regression, which variables were entered as dependent?

9.       In the discussion, it would be useful to know some data on the vaccination situation in the country. for example, what percentage of the population undergoes flu vaccination annually? How many healthcare workers are getting vaccinated?

10.   Overall, the methodological problems of the study place important limitations on the results achieved

Reviewer 3 Report

Comments and Suggestions for Authors

Estimated Authors,

I've read with great interest the present paper focusing on the knowledge, attitudes and practice of a sample of 300 COPD patients from Jordan on Seasonal Flu vaccination (SIV).

The present cross-sectional study suggests that: (a) only a small subset of COPD patient is interested in receiving SIV; (b) higher health literacy and reporting a better attitude are associated with a more positive attitude towards SIV; (c) the most frequently reported reason for not being vaccinated was simply having it forgotten, but a substantial share of respondents claimed some side effects or harmful effects as strictly associated with SIV. 

Collectively, these results are not new, as quite consistent with previous studies on SIV acceptance, but the subset of assessed patients (i.e. COPDs cases) and the region of origin of cases (i.e. middle East) increase the potential significance of these results for the international readers.

In summary, the present reviewer will endorse the acceptance of this study, urging for some minor adjustments.

1) even though the main text is quite appropriate, some minor typos can be noticed, e.g. row 93 to 96, presumptively section "...  ranging from one point for ‘strongly disagree’ to five points for ‘strongly agree’" are duplicated.

2) because of the settings of the study, you should discuss the risk (and eventually explain why you could dimiss it) for some degree of self selection of patients (i.e. patients with higher health literacy are more likely to participate into a survey on health issues);

3) it is unclear whether discussed cases are new diagnoses or consecutive patients admitted to the parent institution during the timeframe of the study; please clarify;

4) could you provide some figures about the acceptance of SIV in the general Jordan population?

Comments on the Quality of English Language

even though the main text is quite appropriate, some minor typos can be noticed, e.g. row 93 to 96, presumptively section "...  ranging from one point for ‘strongly disagree’ to five points for ‘strongly agree’" are duplicated.

Reviewer 4 Report

Comments and Suggestions for Authors

The manuscript presents an overview of COPD patients' perception of the influenza vaccine. It is clear from the data that the low level of education may be responsible for the perception of COPD as a disease and the importance of vaccination. The translated questionnaire and its validation, by the healthcare professional, have to be included in the supplementary file before considering the manuscript for publication. In addition, since the gender is similar despite the fact that most patients with COPD are males, it would be important to stratify each answer depending on the gender. The adherence of the patients to the treatment has to be documented. The amount of passive smokers is quite high, is it more frequent in females? What is the stage of the COPD patients analyzed? How long they have been diagnosed? How many of them have been currently vaccinated against the common cold and influenza? Are there any differences as compared to normal individuals? The point is to ascertain if the individuals are informed of the benefits of vaccination or not.

Some general data on the incidence of the disease worldwide and in the country will benefit the reader. 

The study has its limitations and should be stated. 

There are some grammatical mistakes that should be corrected.

Comments on the Quality of English Language

Some grammatical mistakes were observed in the manuscript

Round 2

Reviewer 2 Report

Comments and Suggestions for Authors

The authors have profoundly modified the manuscript, reshaping the statistics and eliminating some errors that had been reported. In this new version, work can be accepted

Reviewer 4 Report

Comments and Suggestions for Authors

The manuscript was improved in most areas. There are minor details, but as a pilot study can be published.

Comments on the Quality of English Language

Several minor grammatical mistakes were encountered.